# Nuclear Ubiquitin-Proteasome Pathways in Proteostasis Maintenance

**DOI:** 10.3390/biom11010054

**Published:** 2021-01-04

**Authors:** Dina Franić, Klara Zubčić, Mirta Boban

**Affiliations:** Croatian Institute for Brain Research, School of Medicine, University of Zagreb, 10000 Zagreb, Croatia; franicdina@gmail.com (D.F.); klara.zubcic@gmail.com (K.Z.)

**Keywords:** proteasome, ubiquitin, nucleus, inner nuclear membrane, yeast, proteostasis, protein quality control, protein misfolding

## Abstract

Protein homeostasis, or proteostasis, is crucial for the functioning of a cell, as proteins that are mislocalized, present in excessive amounts, or aberrant due to misfolding or other type of damage can be harmful. Proteostasis includes attaining the correct protein structure, localization, and the formation of higher order complexes, and well as the appropriate protein concentrations. Consequences of proteostasis imbalance are evident in a range of neurodegenerative diseases characterized by protein misfolding and aggregation, such as Alzheimer’s, Parkinson’s, and amyotrophic lateral sclerosis. To protect the cell from the accumulation of aberrant proteins, a network of protein quality control (PQC) pathways identifies the substrates and direct them towards refolding or elimination via regulated protein degradation. The main pathway for degradation of misfolded proteins is the ubiquitin-proteasome system. PQC pathways have been first described in the cytoplasm and the endoplasmic reticulum, however, accumulating evidence indicates that the nucleus is an important PQC compartment for ubiquitination and proteasomal degradation of not only nuclear, but also cytoplasmic proteins. In this review, we summarize the nuclear ubiquitin-proteasome pathways involved in proteostasis maintenance in yeast, focusing on inner nuclear membrane-associated degradation (INMAD) and San1-mediated protein quality control.

## 1. Introduction

Maintaining a functional proteome, or proteostasis, is one of the key tasks in the cell. Proteins that are aberrant, for instance due to misfolding, inability to form complexes, or incorrect localization, can be harmful for the cell as a result of a loss of function, interference with other processes or inappropriate interactions with other components in the cell [1,2]. Proteins are at risk of misfolding especially during protein synthesis and assembly into higher-order structures or protein complexes [3]. The effect of protein misfolding and aggregation is particularly evident in proteinopathies, including neurodegenerative diseases such as Alzheimer’s, Parkinson’s, Huntington’s, amyotrophic lateral sclerosis, and others, where accumulation of protein aggregates is a hallmark of pathology. Cells have developed an intricate network of protein quality control (PQC) pathways by which they facilitate folding, assess protein quality, and in the case of an aberrant protein, initiate a proper response to mitigate the damage, either by refolding, or by eliminating the protein via degradation pathways. The main pathway for degradation of misfolded proteins is the ubiquitin-proteasome system.

Degradation-mediated PQC mechanisms have been best described in the cytoplasm and endoplasmic reticulum (ER) [4], however nucleus has emerged as a key PQC compartment for ubiquitination and proteasomal degradation of not only nuclear, but also cytoplasmic proteins [5]. In cells, proteasomes are localized in the cytoplasm, as well as in the nucleus [6,7,8,9]. In fact, in proliferating yeast cells, the majority of cellular proteasomes are localized in the nucleus [10]. Nucleus and the inner nuclear membrane (INM) contain ubiquitination machinery involved in PQC pathways that are important for proteostasis maintenance. In this paper, we summarize recent findings on nuclear ubiquitin-proteasome pathways that function in PQC in *Saccharomyces cerevisiae*: The INM-associated degradation (INMAD) mediated by the E3 ubiquitin ligases Asi1-3, Doa10, and APC/C, and a nuclear pathway for degradation of misfolded proteins mediated by the E3 ubiquitin ligase San1 (Figure 1 and Table 1).

### 1.1. Ubiquitin-Proteasome System

The main site for degradation of misfolded and short-lived proteins is the proteasome, a multiprotein proteolytic machine consisting of a 20S catalytic core particle, and one or two 19S regulatory particles that recognize proteins marked for destruction and regulate substrate entry into the core [11]. Proteins are tagged for proteasomal degradation by attachment of ubiquitin, a small, highly conserved globular protein, which is recognized by ubiquitin binding proteins in the regulatory particle of the proteasome. Regulatory particle additionally contains ATPases that unfold the substrate and translocate it into the 20S chamber for proteolysis by three distinct enzymatic activities, resulting in substrate cleavage into short peptides. The core tunnel is narrow, and as a consequence, proteins must be unfolded prior to degradation by the proteasome [12]. Due to the physical constraints of the 20S core tunnel, the proteasome is able to degrade individual proteins, while protein aggregates and larger structures can be degraded by autophagy [13]. The role of autophagy in degradation of nuclear components has been reviewed elsewhere [14].

Ubiquitin is covalently attached to the protein substrate in a series of enzymatic reactions catalyzed by ubiquitin activating enzyme (E1), ubiquitin-conjugating enzyme (E2), and E3 ubiquitin-protein ligase, usually to the lysine residue side chains [15]. Proteins that have failed to attain a proper structure due to genetic mutations, errors in translation or environmental stress, display degradation signals, or degrons [16], protein segments that are often characterized by exposed stretches of hydrophobic residues. Normally, hydrophobic peptides are buried within the folded protein, localized at the interface with interacting proteins, or embedded within a membrane layer, but may become exposed due to protein misfolding, truncation, or a lack of interaction partner. These signals are recognized by the ubiquitination machinery, often assisted by the molecular chaperones, and the main determinants of protein substrate selectivity are E3 ubiquitin protein ligases [17].

Prior to the delivery to the proteasome, many polyubiquitinated proteins present in protein complexes or embedded within membranes need to be first extracted by the Cdc48 ATPase complex [18]. Cdc48 is a conserved ATP-ase of the AAA+ family (ATPases associated with a variety of cellular activities) whose cellular functions are determined by its association with many different cofactors, including a heterodimer formed by Ufd1 and Npl4 [19]. Within the Cdc48-Ufd1-Npl4 complex, co-factor Npl4 is responsible for recognizing polyubiquitinated substrates, with a strong preference for the K48-type ubiquitin chains [20]. Cdc48-complex bound polyubiquitinated substrate is extracted by passing through the central pore of the Cdc48 homo-hexameric ring, by the ATP hydrolysis, and is subsequently released from the Cdc48-complex.

Polyubiquitinated substrates may be bound by the proteasome directly, by the ubiquitin receptors present in the proteasome regulatory particle. Alternatively, polyubiquitinated substrates could be delivered to the proteasome indirectly, via ubiquitin-like (UbL) and ubiquitin-associated (UBA) family of ubiquitin binding proteins, represented by Dsk2, Rad23, and Ddi1 in yeast [21]. The UBA domain of these proteins binds to ubiquitin on the modified substrates, while their UbL domains bind to ubiquitin receptors at the proteasome regulatory particle [21]. UbL-UBA proteins thus serve as adaptors that link ubiquitinated substrate proteins to the proteasome, delivering them for degradation [21]. The data from a recent study indicate that a large proportion of ubiquitinated proteasome substrates are delivered to the proteasomes indirectly, by UbL-UBA proteins Rad23 and Dsk2 [20].

Substrate-bound polyubiquitin chains can be trimmed or removed by the activity of deubiquitinating enzymes (DUBs), which hydrolyze the bond between substrate and the ubiquitin, or between two ubiquitin molecules [22]. Yeast genome encodes around 20 different DUBs that are localized in ER, mitochondria, nucleus, and cytoplasm, including two DUBs, Ubp6 and Rpn11, that are associated with the proteasome regulatory particle [23]. Different functions of DUBs include protein stabilization or reversal of ubiquitin signaling by the removal of ubiquitin chains from target proteins, editing the ubiquitin modification by trimming the polyubiquitin chains and ubiquitin recycling [22]. A recent screen examining the role of DUBs in protein quality control showed that the degradation of cytosolic quality control substrates is affected by a wide range of the DUB deletion mutants, furthermore the ER-associated degradation is affected by deletion of a DUB gene UBP3, together indicating the involvement of DUBs in the quality control pathways [24].

The ubiquitin-proteasome pathways, including the proteasome, ubiquitination, and deubiquitination machinery, chaperones, and accessory proteins, are conserved from yeast to human [17].

### 1.2. The Nucleus and the Nuclear Envelope at a Glance

The nucleus is enclosed by the nuclear envelope (NE), which consists of two lipid bilayers, the inner and the outer nuclear membrane (INM and ONM) [25]. INM and ONM are joined together at the sites of nuclear pore complexes, which allow nucleocytoplasmic transport. While the ONM is continuous with the ER membrane, the INM has a specific protein composition that considerably differs from that of the ONM and the ER. In metazoan cells the nuclear side of the INM is lined by a meshwork of lamin intermediate filaments and lamin-interacting proteins, called the nuclear lamina [26,27].

Following their co- or post-translational insertion into the ER membrane, integral membrane proteins destined to the INM are able to diffuse from the ER membrane to the ONM and can gain access to the INM via the nuclear pore membrane [28]. Although many mechanistic details of INM protein transport are still unclear, most INM proteins appear to reach the INM by diffusion, with their extraluminal domains passing through the central or lateral channels of the nuclear pore complex and can be retained in the nucleus via interaction with nuclear components, such as lamins or chromatin [28]. INM targeting of certain integral membrane proteins requires active transport, similar to the pathway used by soluble cargo [29].

## 2. Inner Nuclear Membrane-Associated Degradation (INMAD)

### 2.1. Asi1-3 Complex—An Integral Membrane E3 Ubiquitin Ligase at the INM

A new ubiquitination pathway, based on a protein complex formed by two integral membrane E3 ubiquitin ligases, Asi1 and Asi3, has recently been discovered at the INM [30,31]. Asi1 and Asi3 are homologous proteins comprising five membrane-spanning helices, and a C-terminal RING domain oriented towards the nucleoplasm [32,33]. The multimeric Asi-complex additionally contains a third component, an integral membrane protein Asi2, which seems to function as an adaptor that facilitates binding of certain substrates and promotes their ubiquitination by E3 ligases Asi1 and Asi3 [31,34]. Asi1 and Asi3 function together with two E2 enzymes [31], Ubc6, an integral membrane protein, and Ubc7, a soluble protein that is associated with the membrane via integral membrane protein Cue1 [35]. In some cases, Asi1-3 have been reported to function with another combination of E2 enzymes, a soluble E2 enzyme Ubc4, and Ubc7 [30,34]. While Ubc6 and Ubc4 initiate the formation of ubiquitin chains by attaching the first ubiquitin molecule to the substrate, Ubc7 elongates ubiquitin chains by adding additional ubiquitin molecules, mainly via ubiquitin K48 linkage [34,36].

Asi-substrates include integral membrane and soluble proteins (Table 1).

First characterized Asi-substrates were two homologous latent cytoplasmic transcription factors Stp1 and Stp2 [31,37] that function as downstream effectors of the SPS (Ssy1-Ptr3-Ssy5) sensor pathway, a signaling pathway that senses the presence of extracellular amino acids and regulate the expression of specific amino acid permeases [38]. In the absence of inducing amino acids, Stp1/2 are retained in the cytoplasm by the cytoplasmic retention motif RI present within their N-terminal domains [37,38,39]. In the presence of inducing amino acids, the N-terminal domain encompassing RI region is endoproteolytically cleaved, and processed Stp1/2 are able to accumulate in the nucleus where they activate expression of a specific set of genes. However, the cytoplasmic retention mechanism appears to be leaky, as small amounts of full-length Stp1/2 containing the RI region are able to escape and enter nucleus [32]. Intriguingly, in the nucleus RI region also functions as an Asi-dependent degron, which targets unprocessed Stp1/2 that have inappropriately entered nucleus for degradation, thus ensuring Stp1/2 latency [31,37]. The RI motif, comprising around 20 amino acid residues, is predicted to fold into an amphipathic α-helix, a secondary structure characterized by the segregation of hydrophobic and polar residues at two faces of a helix [37]. Considering that the Stp1/2 are soluble proteins, RI-degron is likely recognized by the long nucleoplasmically oriented domains of Asi1-3 or Asi2, or possibly via another, yet unidentified, adaptor.

Integral membrane Asi-substrates include mislocalized integral membrane proteins that normally function in the ER, vacuole, and other membrane compartments in the cell [31,40], and orphan subunits of unassembled protein complexes [34], both of which presumably reach the INM by lateral diffusion from the ER membrane via the nuclear pore membrane. In an assay based on split-GFP complementation system to detect INM-localized integral membrane proteins, around 20 proteins, which are not normally present at the INM in wild-type, were detected at the INM in *asi1Δ* deletion mutant, and additionally, around 40 proteins showed increased INM levels in *asi1Δ* mutant compared to wild-type [40]. In contrast to orphan subunits, INM access of assembled protein complexes may be hindered by the larger size of their cytoplasmic domains, a larger number of transmembrane domains, and possibly interactions with cytosolic components [34].

In comparison to the double mutant *hrd1Δire1Δ*, which has an impaired ERAD (due to the deletion of *HRD1*) and unfolded protein response (due to the deletion of *IRE1*) pathways, the triple mutant *hrd1Δire1Δasi1Δ* that further lacks Asi1 exhibited a severe growth defect when grown at an elevated temperature [31]. The synthetic lethality phenotype demonstrated that Asi1 and Hrd1 function in two distinct parallel pathways, and additionally suggested that Asi complex might be able to target some misfolded proteins that escape ER under the conditions of non-functional ERAD. In support of this possibility, identified Asi-substrates include certain temperature sensitive mutants of integral membrane proteins, which presumably become misfolded at an elevated temperature [34].

How Asi-complex recognizes foreign proteins and distinguishes them from INM resident proteins is not entirely clear. It has been shown that Asi-proteins recognize their membrane substrates via direct interaction with substrate transmembrane domains [34]. Following ubiquitination by Asi1 and Asi3, substrates undergo membrane extraction by the Cdc48-complex [34]. In many cases, efficient degradation of Asi-substrates is dependent on Asi2, however the degradation of some Asi substrates does not require Asi2, indicating that Asi2 may function as a substrate-specific adaptor [30,31]. In accordance with this possibility, Asi1- and Asi3-dependent ubiquitination of certain transmembrane substrates in in vitro reconstituted liposomes was decreased by two thirds when Asi2 was absent [34]. Ubiquitination of Asi2-independent substrates may be based on direct recognition by Asi1 and Asi3 or may involve additional yet unidentified factors.

Although Asi1-3 was able to target some temperature sensitive mutants, which are probably misfolded at a restrictive temperature [34], most of the integral membrane Asi-substrates are wild-type, likely folded proteins that are mislocalized or lack interaction partners. It is possible that foreign proteins are characterized by specific features in their transmembrane domains, which are recognized by direct interaction with Asi1-3, or by a lack of INM-localized or intranuclear interaction partner, which may lead to exposure of Asi-dependent degradation signal. Asi-dependent degradation of soluble unprocessed transcription factor Stp1 containing the RI region could also be categorized as a degradation of mislocalized protein. Cytoplasmic retention determinant RI present within the latent form of Stp1 turns into an Asi-dependent degron when the RI-containing protein enters the nucleus, thus marking RI-containing nuclear proteins as mislocalized. Taken together, Asi-complex seems to primarily function as a scavenger of integral membrane and soluble proteins that are mislocalized to the INM or nucleus, thereby controlling the protein composition at the INM and indirectly regulating nuclear processes, such as gene expression.

### 2.2. Degradation of Nuclear Proteins by the Integral Membrane E3 Ligase Doa10

Doa10 (Degradation Of Alpha2) is an integral membrane E3 ubiquitin ligase in the ER and INM membranes [41,42], best known for its role in ER-associated degradation where it targets integral membrane proteins with lesions in extraluminal or membrane regions, and certain soluble substrates [41,43,44]. In ERAD, Doa10 has a partially overlapping role with the ER-membrane E3 ligase Hrd1 [35], which primarily recognizes substrates with lesions facing the ER lumen and integral membrane substrates [45]. Doa10 and its mammalian homolog Teb4/ MARCHVI [46] comprise multiple membrane-spanning segments and an extraluminally oriented RING domain within its N-terminal region. Doa10 functions with two E2 enzymes, Ubc6 and Ubc7 [41], which exhibit distinct functions in substrate ubiquitination, as described in the previous section [36]. Proteasomal degradation of membrane protein Doa10 substrates additionally requires the Cdc48 complex activity [43].

Nuclear Doa10 substrates include transcriptional repressor MATα2 [41], temperature sensitive mutant of a kinetochore protein Ndc10 [43], and integral INM protein Asi2 [47] (Table 1). Taking into consideration that all proteins are synthesized in the cytoplasm or at the ER-membrane bound ribosomes, Doa10-mediated substrate ubiquitination could occur both in the cytoplasm and/or in the nucleus. Asi2 is only partially stabilized in the *doa10Δ* deletion mutant, indicating an involvement of a parallel pathway in its turnover [47]. Interestingly, lysine-less Asi2 mutant in which all lysine residues have been changed to arginine is also turned over in a Doa10-Ubc6-Ubc7-dependent manner, likely on Ser/Thr residues, indicating the involvement of Doa10-dependent pathway in non-lysine protein ubiquitination [48]. In concert with this finding, it has been shown that Ubc6 is able to attach ubiquitin to hydroxylated side chains of amino acid residues in Doa10-substrates [36]. Another component of the Asi-complex, E3 ligase Asi1 is stabilized in the *ubc7Δ* mutant and in a mutant lacking functional Cdc48 or its adaptor Ubx1, but not in the *doa10Δ* deletion mutant [49]. These findings indicate that distinct pathways are involved in the turnover of specific INM proteins.

Degradation via Doa10 seems to require exposure of hydrophobic regions such as the ones usually buried inside proteins or present at the protein-protein interface [50,51,52]. Doa10-dependent degrons can be cytoplasmic or located within the membrane spanning region of integral membrane proteins [41,43,44,53]. First discovered Doa10 substrate is a short-lived soluble transcriptional repressor MATα2 that contains a Doa10-dependent degradation signal within the first 62 amino acid residues, named Deg1 [41,54]. In fact, Doa10 was identified in a genetic screen looking for doa (Degradation of Alpha2) mutants, using Deg1 degron fused to Ura3, a biosynthetic enzyme that is required for growth on media lacking uracil [41]. The key determinant of the Deg1 degron is the hydrophobic surface of the amphipathic helix [50]. When the degron is hidden within a heterodimer with the transcriptional repressor MATa1, MATα2 protein is stabilized [50]. An amphipathic helix is also a crucial component of a Doa10-dependent degradation signal in the kinetochore protein Ndc10 [53]. In addition to the amphipathic helix, efficient ubiquitination of Ndc10 requires a loosely structured hydrophobic region at the C-terminus [53]. Findings by Ravid and co-workers indicate that Ndc10 degron is normally buried within the protein, however, specific mutations, such as A914T mutation in the temperature sensitive Ndc10-2 mutant, may lead to structural perturbations that expose the degron and thus enable recognition by Doa10, possibly assisted by Hsp70 chaperones [53]. According to structural predictions, N-terminal nucleoplasmically oriented domain of the INM protein Asi2 also possesses a region predicted to form an amphipathic helix (our unpublished data), followed by a region of high hydrophobicity that is predicted to form a transmembrane α helix, which however does not cross the membrane [33]. Based on the analogy to Ndc10 [53], it is possible that these two determinants together form a Doa10-dependent degron.

### 2.3. Regulation of INM SUN-Domain Protein Mps3 Levels via E3 Ligase APC/C-Dependent Pathway

A recent study investigated the degradation of Mps3, a conserved integral SUN-domain protein of the INM that associates with the spindle pole body, the budding yeast equivalent of the vertebrate centrosome [55,56,57]. Spindle pole body is anchored in the nuclear envelope, so that it can simultaneously nucleate both nuclear and cytoplasmic microtubules. A minor fraction of Mps3 is also present throughout the nuclear envelope [58]. Mps3 was targeted for degradation by the proteasome in a Ubc7- and Cdc48-dependent manner, however E3 ligases Asi1-3 and Doa10 known to mediate degradation of other integral membrane proteins at the INM were not involved [59]. Instead, Mps3 degradation required a functional anaphase-promoting complex or cyclosome (APC/C) [58], a conserved E3 ubiquitin ligase important for proper cell cycle progression, which mediates degradation of specific substrates in a precise order in the cell cycle [59,60]. Regulation of the APC/C activity during cell cycle is modulated largely through its association with the co-activators, either Cdc20 or Cdh1. Degradation of Mps3 required APC/C co-activator Cdh1 [58]. Although APC/C does not contain integral membrane components, and its substrates include soluble proteins [59,60], it apparently can gain access to the nuclear periphery where it targets Mps3, possibly via a yet unidentified INM-associated factor. APC/C E3 ligase complex recognizes its substrates by short linear sequence motifs, primarily through so called D and KEN boxes [60]. Nucleoplasmically oriented N-terminal part of Mps3 contains motifs that resemble APC/C-dependent degrons and accordingly the Mps3 N-terminal domain was necessary and sufficient for APC/C^Cdh1^-dependent degradation [58]. The importance of the role of INMAD in proteostasis is highlighted by the phenotype resulting from Mps3 accumulation, which includes an expansion of nuclear membrane and an impaired cell cycle progression [56,58,61].

In conclusion, INMAD pathways are based on the activities of three distinct E3 ubiquitin ligases: (1) integral membrane INM-localized Asi-complex that primarily targets mislocalized proteins and integral membrane orphan complex subunits, (2) integral membrane E3 ligase Doa10, which also localizes to the ER membrane and is best described for its role in degradation of misfolded proteins via ERAD, and (3) nuclear E3 ligase APC/C, which controls the levels of integral membrane SUN-domain protein at the NE. Several issues about the molecular mechanisms of INMAD remain to be explored. An important question is what are the defining features that target integral membrane proteins to INMAD, especially concerning the recognition of the transmembrane domains by the Asi-complex [34]. It is not known how Asi2-independent substrate targeting to the Asi1-Asi3 is mediated, and whether these substrates are recognized directly by Asi1-Asi3, or their recruitment involves additional factors [34]. Furthermore, it is also unknown which Asi2 domains are responsible for the recognition of soluble substrates, such as Stp1/ Stp2. Since Asi2 possesses a long N-terminal nucleoplasmically oriented domain [33], it is plausible that the regions within this domain contribute to the recognition of soluble substrates.

## 3. Nuclear Pathways for Managing Misfolded Proteins

### 3.1. Proteasomal Degradation of Misfolded Proteins via Nuclear San1-Dependent Ubiquitination Pathway

Proteasomal degradation of misfolded and aberrant proteins in the nucleus is primarily mediated by San1, a RING domain E3 ligase localized to the nucleus that preferentially functions with E2 enzyme Ubc1 but can also work with Ubc3/ Cdc34 [5,62,63,64,65].

San1 recognizes its substrates by their exposed hydrophobicity and appears to be particularly selective for misfolded proteins with exposed hydrophobic regions characterized by high insolubility that leads to aggregation [66,67]. San1 substrates include temperature sensitive mutants of nuclear proteins, such as Sir4-9 and Cdc68-1, as well as peptides and truncated proteins [62,66,68,69] (Table 1). Mutations and truncations presumably impair proper folding or cause local misfolding that exposes hydrophobic regions that should normally be buried within the protein [66]. One of the key aspects in selective protein degradation is how ubiquitination machinery distinguishes misfolded proteins from normally folded counterparts. Based on structure prediction analysis, San1 contains natively unstructured domains, which are able to directly bind San1 substrates [70]. Natively unstructured regions can endow a protein with a wide range of distinct conformations, thus providing the ability to bind many differently shaped interaction partners [71]. Indeed, an analysis of San1-substrate interaction using an array of San1 mutants containing small deletions and a number of different substrates revealed that binding of different substrates involved different segments of intrinsically disordered N- and C-terminal San1 regions [70]. In conclusion, flexible, disordered regions enable San1 to directly bind many different misfolded proteins.

Efficient ubiquitination and degradation of certain San1-substrates require Hsp70 chaperones Ssa1 and Ssa2 [72,73], which presumably function upstream of San1, possibly by facilitating the interaction between the substrate and San1. The requirement for Hsp70 chaperones in San1-mediated substrate ubiquitination was in correlation with substrate insolubility, supporting a hypothesis that Hsp70 chaperones maintain the solubility of aggregation-prone substrates in order to enable recognition by San1 [72]. Degradation of certain San1-substrates requires the function of the Cdc48-complex [73,74]. As with Hsp70 chaperones, the requirement for Cdc48 in San1-dependent degradation is in correlation with substrate insolubility [74].

Intriguingly, nuclear San1-dependent pathway targets not only nuclear, but also many misfolded cytoplasmic proteins, such as proteins that were originally constructed as models for studying cytoplasmic protein quality control [73,74,75,76,77,78]. How misfolded cytoplasmic substrates are selected and transported for degradation in the nucleus is not entirely clear. In addition to San1, in many cases efficient degradation of misfolded cytoplasmic substrates requires Ubr1, an E3 ubiquitin ligase best known for its role in N-end rule pathway [79]. Full stabilization of misfolded proteins is often reached only in a double mutant where both San1 and Ubr1 are absent [76,77,78,79,80,81]. Ubr1 was long thought to function solely in cytoplasm, however, the data from a recent study indicate that a large pool of Ubr1 is localized in the nucleus [78]. Available data indicate that Ubr1 mainly targets misfolded proteins in the cytoplasm but can also target substrates in the nucleus.

A recent study suggests that proteasomal targeting of nuclear and cytoplasmic misfolded proteins shows some distinct requirements [77]. While the efficient degradation of cytoplasmic substrates tagged with a nuclear export signal (NES) required modification by mixed K11- and K48-linked ubiquitin chains, degradation of nuclear localization signal (NLS) tagged substrates depended solely on San1-mediated K48-linked ubiquitination. Notably, modification by K11-linked ubiquitin chains was not required for proteasomal degradation of all cytoplasmic substrates in general, such as the N-end rule substrates, which are non-misfolded cytoplasmic substrates, indicating that this is a specific requirement for degradation of misfolded substrates. A more complex proteasome targeting signal involving both K11- and K48-linked ubiquitin chains may provide a stricter regulation of misfolded protein degradation in the cytoplasm, in order to safeguard folding intermediates of newly synthesized proteins from premature degradation [77].

In conclusion, nuclear E3 ligase San1 has a key role in quality control of not only nuclear, but also cytoplasmic misfolded proteins that enter nucleus. In many cases, the efficient degradation of misfolded proteins via San1-pathway requires a collaboration with the E3 ligase Ubr1. San1 can recognize substrates containing exposed hydrophobic stretches directly, via its unstructured N- and C-terminal domains. In some cases, such as with highly insoluble aggregation-prone substrates, San1-mediated ubiquitination is facilitated by Hsp70 chaperones, which presumably promote substrate solubility and assist in substrate binding. Proteasomal delivery of aggregation-prone San1-substrates further requires the activity of a Cdc48 complex.

### 3.2. Managing Ubiquitin-Proteasome System Overload by Sequestration of Misfolded Proteins into Nuclear Inclusions

When the capacity of the ubiquitin-proteasome system is exceeded by the production of misfolded proteins, such as under the conditions of an acute stress, one of the cell strategies to deal with misfolded proteins is sequestration into specialized inclusions in cytoplasm and nucleus that function as a transient storage for later refolding or degradation. In yeast, there are several distinct types of inclusions: perivacuolar insoluble protein deposit (IPOD), cytoplasmic quality control bodies (CytoQ), and intranuclear quality control compartment (INQ) [82,83,84]. Yeast INQ was initially named JUNQ (juxtanuclear quality control compartment), as it was originally described as a structure at the outer side of the nuclear envelope [82]. As it is not yet entirely clear whether INQ and JUNQ are two distinct structures, they are sometimes collectively called JUNQ/INQ [85].

Intranuclear inclusions sequester both nuclear and cytoplasmic proteins [77,83] (Figure 1). The observation that misfolded proteins of cytoplasmic origin accumulate in nuclear and in cytoplasmic inclusions simultaneously, and not sequentially [83], indicates that the nucleus is not merely a backup for an overflow of misfolded proteins from the cytoplasm, but has a key function in mitigating proteotoxic stress. Protein sequestration is an organized process that requires a nuclear chaperone Btn2 and a cytoplasmic chaperone Hsp42, specific aggregases that promote the formation of the inclusions in the nucleus and cytosol, respectively [83,86,87]. The importance of misfolded protein sequestration by Hsp42 and Btn2 is evident in cells with low Hsp70 chaperone capacity, where the sequestration of misfolded protein becomes necessary to prevent proteostasis collapse and to maintain cell viability [88]. The fate of the proteins sequestered into nuclear inclusions can be refolding or degradation, and the outcome seems to be determined depending on the protein disaggregation pathway employed. While Hsp104-dependent disaggregation pathway predominantly targets proteins to refolding pathways, disaggregation via Apj1-dependent pathway results in protein degradation [89].

Quality compartments appear to be conserved from yeast to human, although there are some differences in characteristic features [90]. Intriguingly, in mammalian cells under proteotoxic stress, such as that induced by heat stress or proteasome inhibition, misfolded proteins can be transiently stored in the nucleolus for refolding or degradation [91,92]. Similarly, defective ribosomal products that arise during new protein synthesis were also found to accumulate in nucleoli before clearance by the proteasomes [93]. Together these findings indicate nucleolus as an important QC compartment for managing misfolded and aberrant proteins in mammalian cells.

## 4. SUMO-Targeted Ubiquitin Ligases (STUbLs) in Nuclear Protein Quality Control

SUMO (small ubiquitin-related modifier) is a post-translational protein modifier that belongs to the family of ubiquitin-related proteins [94]. Simple eukaryotic organisms, such as yeast possess a single SUMO gene (*SMT3*), while humans express three SUMO paralogs [94]. Protein modification by SUMO can modulate protein–protein interactions, alter protein conformation or activity, or affect modification by other modifiers, such as ubiquitin [94]. SUMO reversibly modifies thousands of target proteins, the majority of which are nuclear proteins involved in chromatin organization, DNA repair and transcription, as well as protein homeostasis and trafficking [95]. Similar to posttranslational modification by ubiquitin, SUMO is covalently attached to the protein substrates via an isopeptide bond between SUMO C-terminal glycine and the substrate protein lysine side chain, in an enzymatic cascade that requires SUMO-specific E1 activating enzyme, a sole E2 conjugating enzyme Ubc9, and in many cases, one of the SUMO E3 ligases that promote SUMO transfer to the substrate [95]. Proteins can be mono-, multi- or polysumoylated. Sumoylation is a reversible modification, as SUMO can be removed by SUMO-specific proteases.

SUMO-targeted ubiquitin ligases (STUbLs) recognize sumoylated substrates via their SUMO-interacting motifs (SIMs), and act as ubiquitin ligases to modify the sumoylated proteins with the ubiquitin [96], thus regulating their chromatin association or stability [97,98,99,100]. Most STUbL substrates are nuclear proteins, including transcription factors, proteins involved in the DNA repair, and other chromatin-associated proteins [96]. The best characterized STUbLs are the yeast Slx5/Slx8 heterodimer and the mammalian RNF4 and RNF111. A recent study showed that yeast Slx5 and Slx8 are enriched at seven genomic loci termed “ubiquitin hotspots” [101]. Slx5/Slx8 are recruited to the ubiquitin hotspots by the sumoylated transcription factor-like protein Euc1, which is specifically bound to the sequence motif within these sites [101]. Ubiquitin signal at these loci was dependent on Slx5/Slx8, and was enriched in a *cdc48*-mutant, indicating that Cdc48-complex is required to remove ubiquitinated proteins from the ubiquitin hotspots. Besides Euc1, there are likely additional Slx5/Slx8 substrates bound to the ubiquitin-hotspots, which are cleared from DNA by the function of the Cdc48-complex [101].

Modification by SUMO has been shown to modulate the aggregation or inclusion formation by several proteins, some of which are linked to neurodegenerative diseases [102,103,104,105,106,107]. Interestingly, when the aggregation prone fragment of mutant human huntingtin is expressed in yeast cells, Slx5 reduces its nuclear chromatin-associated aggregates, as well as aggregates in the cytosol [108]. STUbL pathway also seems to be to have role in nuclear PQC in the mammalian cells [109,110].

## 5. Concluding Remarks

It has long been known that the majority of the proteasomes in yeast cells are localized in the nucleus [10], indicating that a large part of proteasomal degradation occurs in this compartment. Nuclear accumulation of proteasomes could be linked to degradation of short-lived proteins, many of which are nuclear proteins, such as transcription factors and cell cycle regulators. Moreover, it later became clear that the nucleus does not only degrade nuclear resident proteins but has an important role in for degradation of misfolded proteins from the cytoplasm [73,74,75,76,77]. The role of the nucleus in degradation-mediated PQC has been further broadened by recent discoveries that Asi-mediated INMAD targets mislocalized integral membrane proteins and unassembled protein complexes [30,31,34,40].

Cytoplasmic processes generate proteins with the characteristics of quality control substrates, such as newly synthesized folding intermediates and subunits of unassembled complexes. Spatial separation of protein complex assembly in the ER from degradation of unassembled subunits in the INM may provide more time for the subunits to find their interaction partners, thus facilitating the complex formation [34]. Similarly, confining the processes of protein folding and misfolded protein degradation to different compartments may serve to protect folding intermediates from premature degradation. In support of this notion, misfolded proteins seem to require a more complex ubiquitin tag to mediate degradation in the cytoplasm than in the nucleus, suggesting that an additional layer of control may be necessary to prevent untimely degradation of QC substrates in the cytoplasm [77]. The details how misfolded cytoplasmic substrates are selected and transported for degradation in the nucleus are not entirely clear. One possibility is that these proteins are capable of diffusion or shuttling between the cytoplasm and the nucleus and are caught by ubiquitination machinery either in the cytoplasm, or in the nucleus. This could explain why in many cases both *san1Δ* and *ubr1Δ* deletion mutations are required for complete stabilization of misfolded substrates [76,77,78,79,80]. In support of this possibility, misfolded substrates that were confined to either nucleus or cytoplasm by tagging them with NLS or NES, were targeted by the nuclear and cytoplasmic ubiquitination machinery, respectively [77].

Once modified with the appropriate polyubiquitin tag, proteins are presumably targeted to the proteasomes localized within the same compartment. Beside short-lived nature of these proteins due to degradation in the proteasome, the addition of polyubiquitin chains could considerably increase the size of the substrate, which may impede the passage through the nuclear pore or impose additional requirements for the nuclear import. However, proteasomal inhibition or the inability of the proteasome to recognize the substrate would stabilize the polyubiquitinated proteins, thus their transport across the NE could be envisaged. In line with this possibility, the experiments with mammalian cell lines treated with the proteasome inhibitors showed active transport of polyubiquitinated proteins from the nucleus to the cytoplasm by a nuclear export pathway [111]. In response to proteasome inhibition, K48-linked polyubiquitinated proteins were exported to the cytosol in a CRM1 (exportin-1 [112]) dependent manner, via interaction of polyubiquitinated proteins with a protein complex consisting of the ubiquitin-binding protein UBIN (UBQLN4, [113]) and a NES-containing protein termed polyubiquitinated substrate transporter (POST) [111].

Nuclear PQC-related ubiquitination pathways have been best described in yeast. Asi-proteins and San1 have no apparent mammalian orthologs. However, a complex bioinformatics analysis based on specific San1 features, such as the pattern of intrinsically disordered regions yielded several human E3 ligases with properties that may allow them to recognize substrates using a similar mechanism as San1 [114]. Whether these E3 ligases are functionally equivalent to yeast San1 remains to be investigated.

In the metazoan cells, the nuclear envelope breaks down in each cell division [115], and the proteasomes distribute throughout the cell [6,8], therefore it is conceivable that the proliferating metazoan cells have some distinct characteristics of nuclear quality control compared to differentiated cells or budding yeast with closed mitosis, in which the integrity of the nuclear envelope is preserved. PQC pathways may be especially important in neurons, which are long-lived postmitotic cells that do not have the option to dilute the damage in cell division or get rid of the damage by asymmetric inheritance. Furthermore, as postmitotic cells, neurons do not have the option of mixing of cytoplasmic and nuclear content, as occurs in proliferating cells upon nuclear envelope breakdown, therefore these cells might have a greater reliance on nuclear PQC pathways.

In conclusion, accumulating data indicate nucleus as an important quality control compartment for degradation of aberrant or mislocalized proteins in yeast. Nuclear degradation-mediated PQC pathways include INMAD and San1-dependent degradation. There are no apparent homologues of San1 and Asi-proteins in the mammalian cells, however further research is needed to investigate whether there are functionally equivalent pathways. Findings that the mammalian nucleoli can function as a transient storage of misfolded proteins and defective ribosomal products suggest that the management of aberrant cellular proteins is one of the conserved roles of the nucleus.

**Table 1 biomolecules-11-00054-t001:** Nuclear ubiquitin-proteasome pathways for degradation of protein quality control (PQC) substrates in yeast.

	E3 Ubiquitin Ligase	E2 Involved	Nuclear PQC Substrates		Degradation Signal
**INMAD**					
	Asi1-Asi3	Ubc6 and Ubc7 [31] Ubc4 and Ubc7 [30,34]	nuclear proteins:	Stp1 and Stp2 [31,37]	nuclear RI region (amphipathic helix) [31,37]
			integral INM ^1^ proteins:	INM-mislocalized [31,40]	transmembrane domains [34]
				orphan subunits and lone proteins [34]	
				ts^2^ mutants [34]	
				other [30,31,40]	
	Doa10	Ubc6 and Ubc7 [41]	nuclear proteins:	Matalpha2 [41]	amphipathic helix [50]
				Ndc10-2 (ts) [43]	amphipathic helix and hydrophobic region [53]
			integral INM ^1^ proteins:	Asi2 [47]	not determined
				Mps2-1 (ts) [43]	not determined
	APC/C ^Cdh1^	Ubc7 [58]	integral INM ^1^ protein:	Mps3 [58]	nucleoplasmic KEN-box and D-box-like sequences [58]
**Nuclear**					
	San1	Ubc1 or Ubc3/ Cdc34 [62,63]	nuclear proteins	ts ^2^ mutants of nuclear proteins [62]	exposed hydrophobicity [66]
				truncated proteins [66,70,76]	
				artificial substrates and other mutants [66,67,70,71,72,73,75,76,77]	

^1^ INM—inner nuclear membrane, ^2^ ts—temperature sensitive.

## Figures and Tables

**Figure 1 biomolecules-11-00054-f001:**
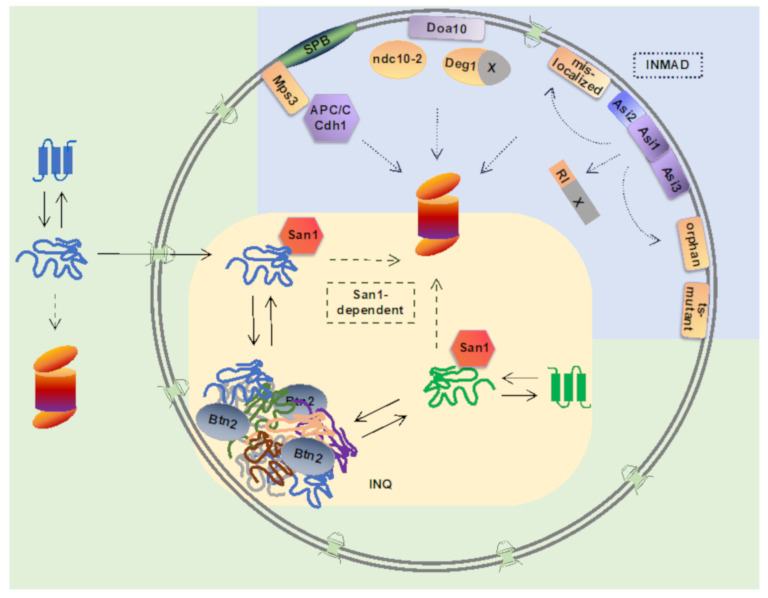
Nuclear ubiquitin-proteasome-dependent protein quality control pathways in yeast. Inner nuclear membrane-associated degradation (INMAD) is mediated via E3 ubiquitin ligases Asi1-3, Doa10, and APC/C. Integral INM-localized Asi-complex consists of E3 ubiquitin ligases Asi1 and Asi3, and a substrate-specific adaptor protein Asi2. Asi-complex targets nuclear RI-degron-containing proteins. Integral membrane Asi-substrates include proteins mislocalized to the INM, orphan subunits of unassembled proteins complexes, and temperature-sensitive (ts) mutants. Integral membrane E3 ligase Doa10 localizes to both the endoplasmic reticulum and the INM, and its substrates include Deg1-degron containing proteins, Ndc10-2 kinetochore mutant protein, and INM protein Asi2. E3 ligase APC/C with its co-activator Cdh1 target Mps3, an integral INM protein of the spindle pole body (SPB). Nuclear E3 ligase San1 targets misfolded cytoplasmic and nuclear proteins for proteasomal degradation. Upon ubiquitin proteasome system overload, misfolded nuclear and cytoplasmic proteins can be reversibly sequestered into intranuclear quality compartment (INQ) by the sequestrase Btn2. Upon disaggregation, misfolded proteins can be directed to refolding or degradation.

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
