# Peer review of "Nuclear Ubiquitin-Proteasome Pathways in Proteostasis Maintenance"

_biomolecules, 2021, doi:10.3390/biom11010054_

Round 1
Reviewer 1 Report
In this manuscript, Franić et al. review a variety of aspects of proteostasis in the nucleus, ranging from the degradation of both membrane and soluble proteins to the sequestration of misfolded proteins in discrete nuclear compartments. The work appears comprehensive and well researched. I enjoyed reading it, in particular the section 3.2 and the concluding remarks.
Research in the field of nuclear PQC is very active and there are many important recent findings which I find the authors could give even more attention to, and conversely consider trimming the sections regarding the early work on Doa10, Ubr1 and San1.
Below I list suggested minor revisions:
Line 18: identifies should be identify
Line 19 and 40: The authors use the term protein “repair”. I don’t think I have come across that before, other than refolding what is done to “repair” a protein?
Line 39: “take proper response” -> “initiate a proper response”
Line 96 and 353: “human” instead of “men” as other genders also have UPS and PQC systems
Line 105-107: please provide a reference for the first paragraph
Line 118: Correct to “ The multimeric Asi1 complex….”
Line 124: It somehow bothers me to read “Ubc6/Ubc4 and Ubc7”. I would either rephrase the sentence to avoid having to write Ubc6/Ubc4. Or delete the whole sentence. The point it makes is very clear from the next sentence
Line 155-157: I find that this paragraph would benefit from a bit of rephrasing. What is written is 100% correct, it's just written in a way that I found confusing and had to re-read a few times.
Line 254: In this paragraph, it would be great if the authors also included a sentence or two with an outlook on what would be the next questions to address in terms of INMAD.
Line 260-327: In this section, 3.1, the authors review the role of San1 in nuclear protein quality control. I suggest that the authors give more weight to the most recent work in this area, and maybe shorten the text that reviews the early San1/nuclear PQC work. In the past years this has already been reviewed frequently; see, e.g. Nielsen et al., 2014; Jones & Gardner, 2016; Enam et al., 2018.
Line 382-383: Here the authors could consider also citing Kriegenburg et al, 2014, Plos Genet.
Author Response
Response to Reviewer 1 Comments
Reviewer 1: In this manuscript, Franić et al. review a variety of aspects of proteostasis in the nucleus, ranging from the degradation of both membrane and soluble proteins to the sequestration of misfolded proteins in discrete nuclear compartments. The work appears comprehensive and well researched. I enjoyed reading it, in particular the section 3.2 and the concluding remarks.
Research in the field of nuclear PQC is very active and there are many important recent findings which I find the authors could give even more attention to, and conversely consider trimming the sections regarding the early work on Doa10, Ubr1 and San1.
Response: We thank Reviewer 1 for the comments and helpful suggestions. We addressed all requested changes, and listed our point-by-point response below. We streamlined the text by trimming some sections, as described below in our point-by-point response.
Below I list suggested minor revisions:
Point 1. Line 18: identifies should be identify
Response 1. We have examined the proposed change in the following sentence:
Line 18: “To protect the cell from the accumulation of aberrant proteins, a network of protein quality control (PQC) pathways identifies the substrates and direct them towards repair or elimination via regulated protein degradation.”
We agree to change the verb “identifies” into the the form “identify” if plural is appropriate. In our understanding, the verb “identify” refers to the noun “network” (singular) and the “PQC pathways” (plural) is a genitive of the noun “network”, therefore, we believe that the verb should have the singular form “identifies”.
Point 2. Line 19 and 40: The authors use the term protein “repair”. I don’t think I have come across that before, other than refolding what is done to “repair” a protein?
Response 2. We changed the term “repair” into “refolding” as suggested, and the sentences are now as following:
Line 19: “To protect the cell from the accumulation of aberrant proteins, a network of protein quality control (PQC) pathways identifies the substrates and direct them towards refolding or elimination via regulated protein degradation.”
Line 41: “Cells have developed an intricate network of protein quality control (PQC) pathways by which they facilitate folding, assess protein quality and in the case of an aberrant protein initiate a proper response to mitigate the damage, either by refolding, or by eliminating the protein via degradation pathways.”
Point 3. Line 39: “take proper response” -> “initiate a proper response”
Response 3. We made the suggested change, and the sentence is now as following:
Line 40: “Cells have developed an intricate network of protein quality control (PQC) pathways by which they facilitate folding, assess protein quality and in the case of an aberrant protein initiate a proper response to mitigate the damage, either by refolding, or by eliminating the protein via degradation pathways.”
Point 4. Line 96 and 353: “human” instead of “men” as other genders also have UPS and PQC systems
Response 4. We made the suggested correction of the term “men”, and changed it into “human”. The sentences are now as following:
Line 110: “The ubiquitin-proteasome pathways, including the proteasome, ubiquitination and deubiquitination machinery, chaperones and accessory proteins, are conserved from yeast to human [17].”
Line 366: “Quality compartments appear to be conserved from yeast to human, although there are some differences in characteristic features [92].”
Point 5. Line 105-107: please provide a reference for the first paragraph
Response 5. We corrected the omission by inserting the reference 28:
Line 123: “Following their co- or post-translational insertion into the ER membrane, integral membrane proteins destined to the INM are able to diffuse from the ER membrane to the ONM, and can gain access to the INM via the nuclear pore membrane [28].”
Point 6. Line 118: Correct to “The multimeric Asi1 complex….”
Response 6. We corrected this omission, and the sentence is now as follows:
Line 134: “The multimeric Asi-complex additionally contains a third component, an integral membrane protein Asi2, which seems to function as an adaptor that facilitates binding of certain substrates and promotes their ubiquitination by E3 ligases Asi1 and Asi3 [31,34].”
Point 7. Line 124: It somehow bothers me to read “Ubc6/Ubc4 and Ubc7”. I would either rephrase the sentence to avoid having to write Ubc6/Ubc4. Or delete the whole sentence. The point it makes is very clear from the next sentence.
Response 7. We agree with the Reviewer that the term “Ubc6/Ubc4” is somewhat confusing. We deleted the sentence as suggested, and the text is now as follows:
Line 140: “In some cases, Asi1-3 have been reported to function with another combination of E2 enzymes, a soluble E2 enzyme Ubc4, and Ubc7 [30,34]. While Ubc6 and Ubc4 initiate the formation of ubiquitin chains by attaching the first ubiquitin molecule to the substrate, Ubc7 elongates ubiquitin chains by adding additional ubiquitin molecules, mainly via ubiquitin K48 linkage [34,36].”
Point 8. Line 155-157: I find that this paragraph would benefit from a bit of rephrasing. What is written is 100% correct, it's just written in a way that I found confusing and had to re-read a few times.
Response 8. We rephrased the paragraph, which now reads as follows:
Line 170-177: “In comparison to the double mutant hrd1Δ ire1Δ, which has an impaired ERAD (due to the deletion of HRD1) and unfolded protein response (due to the deletion of IRE1) pathways, the triple mutant hrd1Δ ire1Δ asi1Δ that further lacks Asi1 exhibited a severe growth defect when grown at an elevated temperature [31]. The synthetic lethality phenotype demonstrated that Asi1 and Hrd1 function in two distinct parallel pathways, and additionally suggested that Asi complex might be able to target some misfolded proteins that escape ER under the conditions of non-functional ERAD. In support of this possibility, identified Asi-substrates include certain temperature sensitive mutants of integral membrane proteins, which presumably become misfolded at an elevated temperature [34].”
Point 9. Line 254: In this paragraph, it would be great if the authors also included a sentence or two with an outlook on what would be the next questions to address in terms of INMAD.
Response 9. We included a short text addressing future perspectives on INMAD research:
Line 274: “Several issues about the molecular mechanisms of INMAD remain to be explored. An important question is what are the defining features that target integral membrane proteins to INMAD, especially concerning the recognition of the transmembrane domains by the Asi-complex [34]. It is not known how Asi2-independent substrate targeting to the Asi1-Asi3 is mediated, and whether these substrates are recognized directly by Asi1-Asi3, or their recruitment involves additional factors [34]. Furthermore, it is also unknown which Asi2 domains are responsible for the recognition of soluble substrates, such as Stp1/ Stp2. Since Asi2 possesses a long N-terminal nucleoplasmically oriented domain [33], it is plausible that the regions within this domain contribute to the recognition of soluble substrates.”
Point 10. Line 260-327: In this section, 3.1, the authors review the role of San1 in nuclear protein quality control. I suggest that the authors give more weight to the most recent work in this area, and maybe shorten the text that reviews the early San1/nuclear PQC work. In the past years this has already been reviewed frequently; see, e.g. Nielsen et al., 2014; Jones & Gardner, 2016; Enam et al., 2018.
Response 10. We have modified the section 3.1., by shortening the description of the early work, referring to the suggested reviews, thus giving more weight to the recent studies.
Lines 286 - 322: “Proteasomal degradation of misfolded and aberrant proteins in the nucleus is primarily mediated by San1, a RING domain E3 ligase localized to the nucleus that preferentially functions with E2 enzyme Ubc1, but can also work with Ubc3/ Cdc34 [63-67].
San1 recognizes its substrates by their exposed hydrophobicity consisting of at least five contiguous hydrophobic amino acids, and appears to be particularly selective for misfolded proteins with exposed hydrophobic regions characterized by high insolubility that leads to aggregation [68,69]. San1 substrates include temperature sensitive mutants of nuclear proteins, such as Sir4-9 and Cdc68-1, as well as peptides and truncated proteins [63,68,70,71] (Table 1). Mutations and truncations presumably impair proper folding or cause local misfolding that exposes hydrophobic regions that should normally be buried within the protein [68]. One of the key aspects in selective protein degradation is how ubiquitination machinery distinguishes misfolded proteins from normally folded counterparts. Based on structure prediction analysis, San1 contains natively unstructured domains at its N- and C-termini, which are able to directly bind San1 substrates [72]. Natively unstructured regions can endow a protein with a wide range of distinct conformations, thus providing the ability to bind many differently shaped interaction partners [73]. Indeed, an analysis of San1-substrate interaction using an array of San1 mutants containing small deletions and a number of different substrates revealed that binding of different substrates involved different segments of intrinsically disordered N- and C-terminal San1 regions [72]. In conclusion, flexible, disordered regions enable San1 to directly bind many different misfolded proteins.
Nevertheless, Efficient ubiquitination and degradation of certain San1-substrates requires Hsp70 chaperones Ssa1 and Ssa2 [74,75]. Mutations in SSA1 and SSA2 did not affect nuclear localization of tested San1 substrates, but their level of ubiquitination was lower, indicating that Hsp70, which presumably function upstream of San1, possibly by facilitating the interaction between the substrate and San1. The requirement for Hsp70 chaperones in San1-mediated substrate ubiquitination was in correlation with substrate insolubility, supporting a hypothesis that Hsp70 chaperones maintain the solubility of aggregation-prone substrates in order to enable recognition by San1 [74]. Degradation of certain San1-substrates further requires the function of the Cdc48-complex [75,76]. Mutants with impaired Cdc48 exhibit elevated levels of ubiquitinated San1-substrates, indicating that Cdc48 functions downstream of San1-mediated ubiquitination [66]. As with Hsp70 chaperones, the requirement for Cdc48 in San1-dependent degradation is in correlation with substrate insolubility [76].
Nuclear San1-dependent pathway targets not only nuclear, but also many misfolded cytoplasmic proteins, such as proteins that were originally constructed as models for studying cytoplasmic protein quality control [75,77–79]. Directing cytoplasmic proteins for ubiquitination and degradation in the nucleus is consistent with mainly nuclear localization of proteasomes in yeast cells [10]. How misfolded cytoplasmic substrates are selected and transported for degradation in the nucleus is not entirely clear. Cytoplasmic substrates of San1 may passively diffuse into the nucleus due to their small size, or be actively imported. Entry of cytoplasmic substrates into the nucleus may be facilitated by chaperones although the exact mechanisms are not clear [67,68,70].
In addition to San1, in many cases efficient degradation of misfolded cytoplasmic substrates requires Ubr1, an E3 ubiquitin ligase best known for its role in N-end rule pathway [81]. Full stabilization of misfolded proteins is often reached only in a double mutant where both San1 and Ubr1 are absent [78–80,82,83]. The role of Ubr1 in ubiquitination of misfolded proteins is distinct from its role in the N-end rule pathway, and has a quality control function, to protect cells from proteotoxic stress. Ubr1 was long thought to function solely in cytoplasm, however, the data from a recent study indicate that a large pool of Ubr1 is localized in the nucleus [80]. Available data indicate that Ubr1 mainly targets misfolded proteins in the cytoplasm, but can also target substrates in the nucleus.”
Point 11. Line 382-383: Here the authors could consider also citing Kriegenburg et al, 2014, Plos Genet.
Response 11. We inserted the citation Kriegenburg et al 2014 into the sentence.
Line 431: “This could explain why in many cases both san1Δ and ubr1Δ deletion mutations are required for complete stabilization of misfolded substrates [78–80,82].”

Reviewer 2 Report
I really enjoyed reading this article - very interesting and well written. I have no further comments.
Author Response
Response to Reviewer 2 Comments
Reviewer 2: No changes required.
Response: We thank Reviewer 2 for the positive feedback.

Reviewer 3 Report
This is an interesting review on ubiquitin-proteasome pathway, focused on nuclear proteostasis. In general, sections are well written and the cited articles are correctly selected. Potentially, it could be an appropriate contribution to the journal. However, a sort of partial or restricted view of the term “ubiquitin-proteasome pathway” is observed.
Important unaddressed aspects:
- There’s an important crosstalk between ubiquitin and sumo in the nucleus, with important impact in protein degradation. A review of the most relevant nuclear roles of sumo signaling intrinsically linked with ubiquitin-proteasome pathway should be included.
- The Cdc48 complex is mentioned in the manuscript due to its important links in protein turnover in the nucleus. Nonetheless, the Cdc48 route is not well explained. The ubiquitin-related Cdc48-associated factors should be described.
- UBL-UBA factors: they are mostly nuclear. Their roles in the pathway should be described, as well.
- DUBs are important regulators of proteostasis, with key roles in the nucleus. They are not even mentioned. This should be corrected as well.
Author Response
Response to Reviewer 3 Comments
Reviewer 3: This is an interesting review on ubiquitin-proteasome pathway, focused on nuclear proteostasis. In general, sections are well written and the cited articles are correctly selected. Potentially, it could be an appropriate contribution to the journal. However, a sort of partial or restricted view of the term “ubiquitin-proteasome pathway” is observed.
Response: We thank Reviewer 3 for the comments. We addressed all requested changes, as listed in our point-by-point response below. We believe that addressing Reviewer’s suggestions resulted in a more complete view of the ubiquitin-proteasome pathway.
Important unaddressed aspects:
Point 1. There’s an important crosstalk between ubiquitin and sumo in the nucleus, with important impact in protein degradation. A review of the most relevant nuclear roles of sumo signaling intrinsically linked with ubiquitin-proteasome pathway should be included.
Response 1. To address the subject of crosstalk between ubiquitin and sumo in the nucleus, we included an additional section “4. SUMO-targeted ubiquitin ligases (STUbLs) in nuclear protein quality control”.
Lines 374 - 406: “SUMO (small ubiquitin-related modifier) is a post-translational protein modifier that belongs to the family of ubiquitin-related proteins [96]. Simple eukaryotic organisms, such as yeast possess a single SUMO gene (SMT3), while humans express three SUMO paralogs [96]. Protein modification by SUMO can modulate protein-protein interactions, alter protein conformation or activity, or affect modification by other modifiers, such as ubiquitin [96]. SUMO reversibly modifies thousands of target proteins, the majority of which are nuclear proteins involved in chromatin organization, DNA repair and transcription, as well as protein homeostasis and trafficking [97]. Similar to posttranslational modification by ubiquitin, SUMO is covalently attached to the protein substrates via an isopeptide bond between SUMO C-terminal glycine and the substrate protein lysine side chain, in an enzymatic cascade that requires SUMO-specific E1 activating enzyme, a sole E2 conjugating enzyme Ubc9, and in many cases, one of the SUMO E3 ligases that promote SUMO transfer to the substrate [97]. Proteins can be mono-, multi- or polysumoylated. Sumoylation is a reversible modification, as SUMO can be removed by SUMO-specific proteases.
SUMO-targeted ubiquitin ligases (STUbLs) recognize sumoylated substrates via their SUMO-interacting motifs (SIMs), and act as ubiquitin ligases to modify the sumoylated proteins with the ubiquitin [98], thus regulating their chromatin association or stability [99-102]. Most STUbL substrates are nuclear proteins, including transcription factors, proteins involved in the DNA repair and other chromatin-associated proteins [98]. The best characterized STUbLs are the yeast Slx5/Slx8 heterodimer and the mammalian RNF4 and RNF111. A recent study showed that yeast Slx5 and Slx8 are enriched at seven genomic loci termed "ubiquitin hotspots" [103]. Slx5/Slx8 are recruited to the ubiquitin hotspots by the sumoylated transcription factor-like protein Euc1, which is specifically bound to the sequence motif within these sites [103]. Ubiquitin signal at these loci was dependent on Slx5/Slx8, and was enriched in a cdc48-mutant, indicating that Cdc48-complex is required to remove ubiquitinated proteins from the ubiquitin hotspots. Besides Euc1, there are likely additional Slx5/Slx8 substrates bound to the ubiquitin-hotspots, which are cleared from DNA by the function of the Cdc48-complex [103].
Modification by SUMO has been shown to modulate the aggregation or inclusion formation by several proteins, some of which are linked to neurodegenerative diseases [104-109]. Interestingly, when the aggregation prone fragment of mutant human huntingtin is expressed in yeast cells, Slx5 reduces its nuclear chromatin-associated aggregates, as well as aggregates in the cytosol [110]. STUbL pathway also seems to be to have role in nuclear PQC in the mammalian cells [101,112].”
Point 2. The Cdc48 complex is mentioned in the manuscript due to its important links in protein turnover in the nucleus. Nonetheless, the Cdc48 route is not well explained. The ubiquitin-related Cdc48-associated factors should be described.
Response 2. We have addressed this point by adding a section on Cdc48, as follows:
Lines 79-87: “Prior to the delivery to the proteasome, many polyubiquitinated proteins present in protein complexes or embedded within membranes need to be first extracted by the Cdc48 ATPase complex [18]. Cdc48 is a conserved ATP-ase of the AAA+ family (ATPases associated with a variety of cellular activities) whose cellular functions are determined by its association with many different cofactors, including a heterodimer formed by Ufd1 and Npl4 [19]. Within the Cdc48-Ufd1-Npl4 complex, co-factor Npl4 is responsible for recognizing polyubiquitinated substrates, with a strong preference for the K48-type ubiquitin chains [20]. Cdc48-complex bound polyubiquitinated substrate is extracted by passing through the central pore of the Cdc48 homo-hexameric ring, by the ATP hydrolysis, and is subsequently released from the Cdc48-complex.”
Due to this change, the following change has been made accordingly:
Line 181: “Following ubiquitination by Asi1 and Asi3, substrates undergo membrane extraction by the Cdc48-complex [34], an AAA-ATPase with protein-unfoldase activity [34].”
Point 3. UBL-UBA factors: they are mostly nuclear. Their roles in the pathway should be described, as well.
Response 3. To address this point, we included an additional paragraph on UBL-UBA proteins, as follows:
Lines 88-97: “Polyubiquitinated substrates may be bound by the proteasome directly, by the ubiquitin receptors present in the proteasome regulatory particle. Alternatively, polyubiquitinated substrates could be delivered to the proteasome indirectly, via ubiquitin-like (UbL) and ubiquitin-associated (UBA) family of ubiquitin binding proteins, represented by Dsk2, Rad23 and Ddi1 in yeast [21]. The UBA domain of these proteins binds to ubiquitin on the modified substrates, while their UbL domains bind to ubiquitin receptors at the proteasome regulatory particle [21]. UbL-UBA proteins thus serve as adaptors that link ubiquitinated substrate proteins to the proteasome, delivering them for degradation [21]. The data from a recent study indicate that a large proportion of ubiquitinated proteasome substrates are delivered to the proteasomes indirectly, by UbL-UBA proteins Rad23 and Dsk2 [20].”
Point 4. DUBs are important regulators of proteostasis, with key roles in the nucleus. They are not even mentioned. This should be corrected as well.
Response 4. To address this point, we included an additional paragraph on DUBs, as follows.
Lines 98-111: “Substrate-bound polyubiquitin chains can be trimmed or removed by the activity of deubiquitinating enzymes (DUBs), which hydrolyze the bond between substrate and the ubiquitin, or between two ubiquitin molecules [22]. Yeast genome encodes around twenty different DUBs that are localized in ER, mitochondria, nucleus and cytoplasm, including two DUBs, Ubp6 and Rpn11, that are associated with the proteasome regulatory particle [23]. Different functions of DUBs include protein stabilization or reversal of ubiquitin signaling by the removal of ubiquitin chains from target proteins, editing the ubiquitin modification by trimming the polyubiquitin chains and ubiquitin recycling [22]. A recent screen examining the role of DUBs in protein quality control showed that the degradation of cytosolic quality control substrates is affected by a wide range of the DUB deletion mutants, furthermore the ER-associated degradation is affected by deletion of a DUB gene UBP3, together indicating the involvement of DUBs in the quality control pathways [24].”
Additionally, we added the word “deubiquitination” into the following sentence:
Line 111: “The ubiquitin-proteasome pathways, including the proteasome, ubiquitination and deubiquitination machinery, chaperones and accessory proteins, are conserved from yeast to human.”

Round 2
Reviewer 3 Report
The authors addressed all the points and the manuscript should be accepted for publication.